# Responses to Different Magnesium Supply Treatments in the Mature Leaves of *Cunninghamia lanceolata* Seedlings: Morphological, Physiological, and Structural Perspectives

**DOI:** 10.3390/plants14223542

**Published:** 2025-11-20

**Authors:** Yaling Zhang, Bigui Su, Sheng Lu, Tianran Han, Fenglin Wang, Guochang Ding, Chao Wu, Guangqiu Cao, Yu Chen

**Affiliations:** 1College of Forestry, Fujian Agriculture and Forestry University, Fuzhou 350002, China; 52404022076@fafu.edu.cn (Y.Z.); 52504022083@fafu.edu.cn (B.S.); 52404022025@fafu.edu.cn (S.L.); 12504055006@fafu.edu.cn (T.H.); 2Key Laboratory for Forest Adversity Physiological Ecology and Molecular Biology, Fuzhou 350002, China; pinapplejiang@126.com (F.W.); fjdgc@fafu.edu.cn (G.D.); cncgq@126.com (G.C.); 3College of Computer and Information Science, Fujian Agriculture and Forestry University, Fuzhou 350002, China; chaowu@fafu.edu.cn

**Keywords:** *Cunninghamia lanceolata*, magnesium stress, photosynthesis, chloroplast ultrastructure, plantation management

## Abstract

(1) Background: Repeated planting cycles and monoculture practices have led to widespread magnesium (Mg) deficiency in Chinese fir (*Cunninghamia lanceolata*) plantations. To gain clarity on how different Mg concentrations affect seedling growth and physiology, we designed the following experiment. (2) Methods: One-year-old seedlings were exposed to three Mg concentration treatments: High (HM), Medium (MM), and Low (LM). Their responses were evaluated in terms of growth traits, photosynthetic activity, and chloroplast structure. (3) Results: Both HM and LM significantly affected leaf development, with LM having the strongest impact. LM disrupted chloroplast structure, causing thylakoid membrane rupture, mitochondrial damage, accumulation of osmiophilic granules, and increased spacing between chloroplasts and cell walls. LM also impaired photosynthesis, lowering the net photosynthetic rate (Pn) and peroxidase (POD) activity, while increasing malondialdehyde (MDA) levels. Leaf growth was reduced, as shown by smaller leaf area and lower biomass. In contrast, HM temporarily enhanced some physiological traits, including intercellular CO_2_ concentration (Ci), transpiration rate (Tr), leaf dry matter content (LDMC), and ATPase activity, though it also reduced Fv/Fo compared to MM. (4) Conclusions: Both high and low Mg concentration negatively affected photosynthesis, with Mg deficiency causing the most severe damage. These findings highlight the importance of managing soil Mg levels to maintain healthy growth and productivity in *C. lanceolata* plantations.

## 1. Introduction

Photosynthesis is the foundation of plant productivity, supplying energy for most life on Earth [1]. As the primary site of photosynthesis, leaves are highly responsive to environmental conditions [2,3]. Their efficiency depends on chloroplast integrity, enzyme activity, and pigment content-factors closely linked to soil nutrient availability [4,5,6]. Among these nutrients, magnesium (Mg) plays a critical role. It is the central atom in chlorophyll and is essential for enzyme activation, membrane and ribosome stability, carbon and nitrogen metabolism, and the regulation of oxidative stress [7].

Chinese fir (*Cunninghamia lanceolata* (Lamb.) Hook.), a fast-growing conifer native to southern China, is a key species in regional plantation forestry [8] due to its economic value and adaptability [9,10]. However, its biology and management practices contribute to nutrient imbalances. Old leaves decompose slowly, limiting nutrient return to the soil [11]. Additionally, long-term monoculture and repeated rotations have caused ongoing nutrient depletion, including Mg loss [12,13], leading to declines in soil fertility and plantation productivity [14,15]. Research indicates that environmental stressors like high temperatures [16] and acid rain [17] further increase Mg leaching. Because Mg is mobile in soil [18] and competes with other ions [19,20], its effective availability is often low. Yet most fertilization regimes still prioritize nitrogen, phosphorus, and potassium, while Mg remains underutilized [21,22]. Targeted Mg management could improve photosynthetic performance and help restore productivity in nutrient-poor *C. lanceolata* stands [23,24].

From a plant developmental biology perspective, leaf development is co-regulated by multiple factors, including environmental conditions, nutrient status, hormones, and genetic control [25,26]. It typically follows a stage-specific developmental trajectory that progresses from primordium initiation and polarity establishment to subsequent expansion [27]. As leaves mature, their photosynthetic capacity gradually increases until reaching a stable level [28]. Previous studies have systematically elucidated the developmental processes of species such as *Nicotiana tabacum* (tobacco) [29], *Ginkgo biloba* (ginkgo) [30], and *Areca catechu* (areca palm) [31]. However, research on *C. lanceolata* has predominantly focused on individual growth stages or isolated physiological and biochemical indicators [32,33]. To date, no study has comprehensively tracked the continuous dynamic changes in morphological development, physiological metabolism, and cellular structure of *C. lanceolata* throughout the entire process from bud germination to full leaf maturation [34]. This lack of systematic investigation has greatly constrained a holistic understanding of the regulatory mechanisms underlying leaf development in *C. lanceolata*.

As a core functional element of the plant photosynthetic system, Mg not only serves as the central atom of chlorophyll but also plays an essential role in key physiological processes such as photosynthetic electron transport and carbon assimilation [35,36]. Numerous studies have demonstrated that Mg deficiency impairs photosynthetic performance and overall plant vitality. Therefore, the scientific regulation of nutrient supply to optimize leaf photosynthetic characteristics represents an important strategy for overcoming obstacles associated with continuous cropping [37,38,39,40]. In this context, the present study employed *C. lanceolata* seedlings as experimental materials to systematically investigate the multi-level response mechanisms-from structural to physiological to phenotypic levels-of *C. lanceolata* under different Mg concentration treatments.

First, by observing chloroplast ultrastructure through transmission electron microscopy, we directly examined how varying Mg concentrations affect the integrity and stability of core chloroplast components (e.g., thylakoid membranes), thereby providing a cytological basis for analyzing changes in photosynthetic function. Next, the degree of Mg stress–induced inhibition of photosynthetic capacity was evaluated by determining leaf photosynthetic parameters and chlorophyll fluorescence characteristics. Meanwhile, through the assessment of antioxidant system activity and key photosynthetic enzyme activity, the underlying physiological mechanisms mediating these responses were further elucidated. Finally, the comprehensive effects of Mg stress on the growth and development of *C. lanceolata* were quantified by measuring leaf mineral element content and growth indicators such as leaf area.

Based on these research objectives, two core hypotheses were proposed in this study: (1) Low Mg concentrations disrupt the photosynthetic system at multiple levels, including chloroplast structure, enzyme activity, and photosynthetic physiological processes; and (2) both high and low magnesium concentration treatments disturb the normal growth pattern of seedlings, triggering adaptive regulatory mechanisms that enable them to cope with environmental stress. The findings of this study provide a theoretical foundation for precision Mg fertilization in *C. lanceolata* plantations in Mg-deficient regions and hold significant practical implications for promoting the sustainable management of *C. lanceolata* forests [39].

## 2. Materials and Methods

### 2.1. Plant Materials and Experimental Site

The experiment was conducted in an artificial climate chamber at Fujian Agriculture and Forestry University where is located in Fuzhou, Fujian Province, China (119°27′ E, 26°09′ N). One-year-old clonal seedlings (clone 020) of *C. lanceolata* were sourced from the Yangkou State-owned Forest Farm in Shunchang County, Fujian Province. In March 2023, healthy seedlings with uniform growth and no visible pest or disease symptoms were selected. On average, seedlings measured 40.8 ± 1.3 cm in height, with a basal diameter of 0.44 ± 0.03 cm and a crown width of 16.7 ± 1.6 cm. Seedlings were transplanted into plastic pots (19 cm top diameter, 15.5 cm bottom diameter, 20 cm height) filled with a sterilized mixture of quartz sand and perlite (3:1, *v*/*v*; sieved to 1 mm). Transplanting occurred one month prior to treatment to allow for acclimation in a growth chamber. Environmental conditions were maintained at a photosynthetic photon flux intensity of 100–110 μmol·m^−2^·s^−1^, relative humidity of 65–70%, and temperature between 26 and 30 °C. During the acclimation period, seedlings were irrigated with a complete nutrient solution as described in Pérez-Harguindeguy’s study [41]. The nutrient solution contained macronutrients (2.5 mM KNO_3_, 2.5 mM Ca(NO_3_)_2_, 0.5 mM KH_2_PO_4_, 1 mM MgSO_4_) and micronutrients (10 μM H_3_BO_3_, 2 μM MnCl_2_, 2 μM ZnSO_4_, 0.5 μM CuSO_4_, 0.065 μM (NH_4_)_6_Mo_7_O_24_, 20 μM Fe-EDTA). To prevent salt accumulation, roots were flushed with deionized water every 15 days.

### 2.2. Different Magnesium Concentration Treatments

Following acclimation, seedlings were rinsed with deionized water and randomly assigned to one of three magnesium (Mg) treatments: low Mg concentration, medium Mg concentration, and high Mg concentration. Each treatment group included 20 seedlings, for a total of 60. The three Mg treatments were implemented by adjusting the concentration of MgSO_4_ in the nutrient solution, while all other components of the nutrient solution remained constant. Specifically, the complete nutrient solution used for all treatments contained macronutrients (2.5 mM KNO_3_, 2.5 mM Ca(NO_3_)_2_, 0.5 mM KH_2_PO_4_) and micronutrients (10 μM H_3_BO_3_, 2 μM MnCl_2_, 2 μM ZnSO_4_, 0.5 μM CuSO_4_, 0.065 μM (NH_4_)_6_Mo_7_O_24_, 20 μM Fe-EDTA). For the low Mg concentration treatment (LM), no MgSO_4_ was added, and 1 mM Na_2_SO_4_ was included to maintain ion balance. For the medium Mg concentration treatment (MM), 1 mM MgSO_4_ was added. For the high Mg concentration treatment (HM), 3 mM MgSO_4_ was added. Seedlings were irrigated with 100 mL of the designated nutrient solution every two days and leached with deionized water every 15 days to prevent salt accumulation.

### 2.3. Sample Collection

Leaf physiological parameters were assayed at 30, 60, 90, and 120 days after treatment (DAT). Sampling positions were standardized by selecting the third whorl of lateral branches below the shoot apex. For each plant, one lateral branch was collected at each time point, with different branches used across time points to avoid excessive damage and ensure data reliability. If fewer than four lateral branches were present in the designated whorl, subsequent measurements were conducted by randomly selecting branches within the same whorl. From each selected lateral branch, twelve leaves were randomly collected and combined into three composite samples (four leaves per composite). Each composite sample was analyzed as one technical replicate, and the mean of the three replicates represented the physiological value for that plant. For each treatment group, twenty plants were independently sampled according to this protocol, yielding twenty biological replicates in total.

Prior to leaf harvesting, non-destructive measurements were performed on the pre-selected leaves, including the determination of photosynthetic parameters and the detection of chlorophyll fluorescence. Upon completion of these assessments, the target leaves were harvested for subsequent analyses. These analyses included the quantification of mineral element contents, the assay of antioxidant and photosynthetic enzyme activities, and the ultrastructural observation of chloroplasts via microtome sectioning.

### 2.4. Leaf Morphology and Biomass Measurement

Leaves were soaked in distilled water at 5 °C in the dark for 12 h to achieve full turgor. After gently blotting surface moisture with absorbent paper, saturated fresh weight and dry weight were measured using an analytical balance (CP224C, ±0.1 mg, Ohaus, Parsippany, NJ, USA). Leaf area was calculated using Image-Pro Plus 6.0. Samples were then oven-dried to a constant weight. Specific leaf area (SLA) and leaf dry matter content (LDMC) were calculated as follows:SLA = leaf area/leaf dry weightLDMC = leaf dry weight/saturated fresh weight

### 2.5. Determination of Mineral Elements in Leaves

Leaves were oven-dried (105 °C for 30 min, followed by 75 °C until constant weight), ground, and sieved through a 0.149 mm mesh. Total carbon (C) and nitrogen (N) were analyzed with an elemental analyzer (Vario Macro Cube). Phosphorus (P), potassium (K), calcium (Ca), magnesium (Mg), manganese (Mn), copper (Cu), zinc (Zn), iron (Fe), and aluminum (Al) were measured using inductively coupled plasma atomic emission spectrometry (ICP-AES; Perkin ICP-MS 8000, PerkinElmer, Waltham, MA, USA) after H_2_SO_4_-HClO_4_ digestion.

### 2.6. Photosynthetic Physiology Analysis

Photosynthetic pigments [42]: Fresh leaves were extracted in 95% ethanol in darkness until fully decolorized. Extracts were centrifuged at 4000 rpm for 10 min at 4 °C, and absorbance was recorded using a UV-Vis spectrophotometer at 645, 663, and 470 nm to quantify chlorophyll a, chlorophyll b, and carotenoids, respectively.

Chlorophyll fluorescence: Fluorescence measurements were taken between 9:00 and 11:00 a.m. on sunny days. After 30 min of dark adaptation, using a portable fluorometer (Pocket-PEA, Hansatech, UK). Parameters recorded included Fo, Fm, Fv, Fv/Fm, and Fv/Fo.

Gas exchange: The measurements were conducted from 9:00 a.m. to 11:00 a.m. on sunny days. Net photosynthetic rate (Pn), intercellular CO_2_ concentration (Ci), transpiration rate (Tr), and water-use efficiency (WUE = Pn/Tr) were measured with a LI-6400XT portable photosynthesis system under ambient CO_2_ concentration (400 μmol·mol^−1^). Photosynthetic photon flux density (PPFD) was adjusted to ambient light, with automatic control of airflow.

### 2.7. Oxidation-Related Indicators and Photosynthetic Enzyme Activities Analysis

The determinations of malondialdehyde (MDA) and peroxidase (POD) activity were performed following the methods described by Xudan Zhou et al. [43].

Lipid peroxidation: MDA content was determined using the thiobarbituric acid (TBA) method. Fresh leaves were ground with SiO_2_ and 10% trichloroacetic acid, reacted with 0.6% TBA at 100 °C for 15 min, and absorbance was measured at 532 nm.

Antioxidant enzymes: POD activity was measured using the guaiacol method. Fresh leaves were homogenized in phosphate buffer and centrifuged. The reaction mixture containing the enzyme extract, guaiacol, and H_2_O_2_ was prepared, and absorbance was recorded at 470 nm to determine POD activity.

Enzyme activities: Rubisco and Ca^2+^Mg^2+^-ATPase activities were quantified using commercial assay kits (Komin Biotechnology, Suzhou, China), following the manufacturer’s protocols.

### 2.8. Chloroplast Ultrastructure Observation

First, fixation was performed: leaf tissue cubes (<1 mm^3^ in size) were first fixed in 2.5% glutaraldehyde prepared in phosphate-buffered saline (PBS) for 6 h, followed by three 15 min washes in 0.1 mol/L phosphate buffer. Subsequently, the samples were fixed in 2% osmium tetroxide for 2 h and rinsed three more times with the same 0.1 mol/L phosphate buffer, for 15 min per rinse. Next, dehydration was carried out: samples were sequentially dehydrated in a 4 °C refrigerator via immersion in 50% ethanol, 70% ethanol, 90% ethanol, a 1:1 mixture of 90% ethanol and 90% acetone, and 90% acetone, with 15–20 min per step, and after this gradient dehydration, they were rinsed three times with 100% acetone at room temperature, 15 min per rinse. Following dehydration, embedding was conducted: samples were first incubated at room temperature for 3–4 h in a 3:1 pure acetone-embedding medium mixture, then incubated overnight at room temperature in a 1:1 pure acetone-embedding medium mixture, followed by additional incubation in a 1:3 acetone–embedding medium mixture at room temperature, and finally treated with pure embedding medium three times at 37 °C, 2–3 h per treatment. Subsequently, polymerization was completed by first placing the samples in a 37 °C oven overnight, then moving them to a 45 °C oven for 12 h, and ultimately polymerizing them at 60 °C for 24 h. Finally, ultrathin sections (50–60 nm thick) were cut using an ultramicrotome, stained with a double protocol (3% uranyl acetate followed by lead citrate), and observed under a transmission electron microscope (FEI Tecnai Spirit G2 BioTWIN, FEI, Hillsboro, OR, USA).

### 2.9. Data Analysis and Structural Equation Model Construction

Data were analyzed using one-way analysis of variance (ANOVA), followed by Duncan’s multiple range test (*p* < 0.05). All statistical analyses were performed in SPSS Statistics 26.0. To evaluate the importance of each measurement indicator for constructing the structural equation model, an out-of-bag scoring random forest model was trained based on the final (120 days) and initial (30 days) values of the selected indicators. For each feature, 30 permutations were performed to calculate permutation importance analysis, assessing the extent of model performance degradation after permutation. SmartPLS v3.2.9 was used to construct the PLS-SEM model based on the residuals between the final and initial values of the selected features.

## 3. Results

### 3.1. Effects of Magnesium Treatments on Leaf Morphology and Chemical Traits

Specific leaf area (SLA) declined over time under all treatments. Both the MM and HM groups showed an initial increase followed by a decrease, with SLA values in HM consistently lower than those in MM. In contrast, SLA in the LM declined steadily throughout the study. Although SLA in LM was significantly higher than the other treatments at 30 days, it became markedly lower by 60 days (Figure 1a). Leaf dry matter content (LDMC) increased progressively under all treatments. HM showed the highest LDMC, with significant differences emerging at 90 days. LM also exhibited higher LDMC than MM between 30 and 90 days, but by 120 days, no significant differences were observed among the three treatments (Figure 1b). Leaf area in both MM and LM followed a general pattern of increase, decline, and partial recovery. However, LM consistently had smaller leaf area than MM throughout. In contrast, HM showed a continuous increase followed by a decline, with a later peak than the other groups. At 90 days, leaf area in HM was significantly greater than in MM and LM (Figure 1c). Leaf dry weight increased steadily under all treatments. LM remained consistently lower than MM across all time points (Figure 1d).

Leaf nutrient concentrations shifted in response to magnesium availability. Nitrogen levels were higher in LM at 60 days but fell below those in MM and HM by 90 days (Figure 2a). Carbon concentrations were higher in HM during the early stages and significantly lower in LM at 60 days (Figure 2b). Magnesium content closely followed treatment levels, consistently ranking as HM > MM > LM (Figure 2d). Potassium displayed a decline followed by a rebound, with HM significantly lower than MM and LM in later stages (Figure 2e). Calcium increased over time across all treatments, though the rate of increase was slower in HM and LM (Figure 2f). Aluminum remained higher in MM, while HM showed significantly lower levels at 90 days and LM at 120 days (Figure 2h). Phosphorus and copper both exhibited overall declines, although copper exhibited a late-stage rebound in MM (Figure 2c,k). Iron, zinc, and manganese each declined initially before partially recovering, with MM maintaining slightly higher concentrations at the end of the experiment (Figure 2g,i,j).

### 3.2. Magnesium-Mediated Changes in Photosynthetic Performance and Chlorophyll Metabolism

Chlorophyll a (Chl a), chlorophyll b (Chl b), and carotenoids (Car) all followed a similar pattern, increasing during early development and declining later. Concentrations were lowest at 30 days and peaked between 60 and 90 days. At these peak stages, LM had significantly lower Chl a and Chl b levels compared to MM and HM. By 120 days, Chl a and Chl b concentrations in HM had declined below those in MM, while Car levels in LM were significantly higher than in the other treatments (Figure 3a–c). The Chl a/b ratio initially declined and later recovered, with a significant difference observed only at 60 days, when LM was lower than MM (Figure 3d).

Net photosynthetic rate (Pn) peaked at 30 days across all treatments before declining. HM maintained higher Pn values than MM and LM up to 60 days, but by 120 days, Pn in LM had dropped significantly below that in MM (Figure 4a). Intercellular CO_2_ concentration (Ci) showed distinct patterns: HM declined steadily over time, MM peaked at 60 days before decreasing, and LM rose initially before falling (Figure 4b). Transpiration rate (Tr) followed a trend similar to Pn, peaking at 30 days. MM exhibited the lowest Tr at 90 days before recovering (Figure 4c). At 30 days, Gs increases with the rise in magnesium concentration. As the number of days of growth and development changes, Gs under HM and LM treatments gradually decreases. Under MM treatment, a trough appears at 90 days (Figure 4d). Water-use efficiency (WUE) peaked at 90 days under all treatments, with LM recording the lowest efficiency at that point (Figure 4e).

Chlorophyll fluorescence parameters responded to changes in magnesium supply. Both minimal (Fo) and maximal (Fm) fluorescence increased initially, peaking between 60 and 90 days, before declining (Figure 5a,b). The Fv/Fm ratio reached its lowest point at 90 days, but exhibited no clear treatment-related differences except at 60 days, when the HM group was significantly lower (Figure 5c). A similar trend was observed in the Fv/Fo ratio, with HM falling significantly below MM at 60 days (Figure 5d).

### 3.3. Effects on Enzyme Activities: Carbon Assimilation and Oxidative Stress

Malondialdehyde (MDA), a marker of oxidative stress, showed divergent trends across treatments. Levels in HM and MM peaked mid-period before declining, whereas LM fluctuated more strongly and peaked at 90 days. From 90 to 120 days, MDA levels in LM remained significantly higher than in HM and MM (Figure 6a). Peroxidase (POD) activity increased steadily in HM and MM but varied irregularly in LM, where it declined over time. MM consistently showed the highest POD activity (Figure 6b). Rubisco activity followed a rise-and-fall pattern across treatments. LM displayed marked fluctuations and differed significantly from HM and MM for most of the study period (Figure 6c). Ca^2+^Mg^2+^-ATPase activity varied sharply among treatments. At 30 days, activity was highest in HM, followed by LM and MM. LM peaked at 60 days before dropping sharply, while HM surpassed both other treatments at 90 and 120 days. Statistical analysis confirmed significant differences among treatments at most time points (Figure 6d).

### 3.4. Magnesium-Induced Changes in Chloroplast Ultrastructure

Transmission electron microscopy revealed progressively distinct ultrastructural differences among treatments (Figure 7). By 30–60 days, all groups had developed basic chloroplast structures. However, LM already showed irregular cell walls and loosely arranged chloroplasts, in contrast to the more organized structures in HM and MM. By 90 days, these differences became more pronounced. HM retained dense thylakoids and abundant osmiophilic granules, while MM exhibited well-organized thylakoids with few starch grains. In LM, chloroplasts appeared irregular, partially detached from the cell wall, with visible cavities. At 120 days, HM and MM maintained smooth cell walls and compact thylakoid organization, with HM showing a higher accumulation of osmiophilic granules. In contrast, LM displayed distorted cell walls, fragmented thylakoids, enlarged gaps between chloroplasts and the wall, and signs of mitochondrial degradation.

### 3.5. Key Factors Regulating Mature Leaves Growth Mediates by Different Magnesium Supply

An independent random forest analysis was performed to further identify the potential factors regulating mature leaf growth under varying levels of magnesium (Mg) supply. The analysis revealed that five variables, including Zn, Fv, Chla and et al., exhibited relatively small mean permutation errors (<0.0), suggesting weak associations with variations in mature leaf growth. In contrast, the remaining 19 variables were identified as potential influential factors (Figure 8a). Based on the regression residuals between the final and initial measurements, two partial least squares path modeling (PLS-PM) frameworks were subsequently constructed using the variables selected from the random forest analysis. The photosynthesis-related model (Figure 8b) indicated that both high (HM) and low (LM) Mg supplies indirectly affected the activity of key photosynthetic enzymes by modulating oxidative/antioxidative physiological responses and photosynthetic performance. Among all variables, MDA content showed the highest relative importance. Notably, the two key photosynthetic enzymes, Ca^2+^/Mg^2+^-ATPase and Rubisco, exerted significant but opposite effects on the dry weight (DW) of mature leaves. In the nutrient-related model (Figure 8c), mobile nutrients (N, P, and K) and Ca content were identified as potential determinants of mature leaf growth. Moreover, MDA content mediated the effect of Mg supply on leaf aluminum (Al) content, which subsequently influenced DW through its impact on Ca content. It should be emphasized that the overall performance of both structural equation models was acceptable but not optimal, with GoF values of 0.39 and 0.34, and SRMR values of 0.093 and 0.109, respectively. The integration of the two models proved difficult, likely due to the inherent differences between the two categories of variables. Specifically, photosynthetic parameters represent the instantaneous physiological state of the photosynthetic apparatus, whereas nutrient parameters reflect longer-term processes such as absorption, translocation, accumulation, and remobilization.

## 4. Discussion

### 4.1. Dynamic Changes in Leaf Functional Traits During Development

Leaves are the main photosynthetic organ, and their development reflects the plant’s capacity to assimilate carbon. As leaves mature, specific leaf area (SLA) typically decreases [44], signaling a shift from rapid expansion to structural stability [45]. During this transition, antioxidant enzymes such as peroxidase (POD) help protect developing organelles by stabilizing thylakoid membranes [46,47].

Magnesium (Mg) plays a central role in regulating leaf traits in plants [48]. It supports chlorophyll synthesis and boosts photosynthetic efficiency [49], promoting both leaf expansion and pigment accumulation [50]. Mg also activates essential enzymes, including Rubisco [51] and Ca^2+^/Mg^2+^-ATPase, which drive carbohydrate metabolism, energy transfer, and photosynthate transport [52]. In addition, Mg helps maintain POD activity and limits malondialdehyde (MDA) buildup, preserving membrane integrity [53,54]. It also stabilizes chloroplast structure, preventing thylakoid damage and sustaining photosynthetic function [55,56,57]. Altogether, these functions indicate that adequate Mg enhances structural stability, photosynthetic performance, and stress resistance [58].

### 4.2. Responses of C. lanceolata Leaves to Magnesium Stress

As a core component of chlorophyll [59], Mg directly influences enzyme activity, thylakoid stacking, fluorescence efficiency, photosystem II (PSII) function, carbon fixation, and electron transport [60]. Previous studies have shown that during leaf development, traits shift from acquisitive to conservative strategies [61], with corresponding adjustments in enzyme activity, nutrient allocation, and photosynthetic capacity. Understanding how leaves respond to Mg stress supports forest productivity, ecological restoration, and the management of *C. lanceolata* [62].

Low magnesium concentration treatments (LM), photosynthetic function is severely impaired. Chloroplast ultrastructure is disrupted [63], with fragmented thylakoids, damaged mitochondria, increased plastoglobuli, and widened gaps between chloroplasts and cell walls [64], all of which reduce light capture and electron transport efficiency. LM also disrupts PSII reaction centers and PSI electron flow [65], lowering photochemical conversion and altering gas exchange [66]. Pn drops sharply, Fv/Fm falls below 0.8 [67], and Fv/Fo remains consistently low [68]—clear signs of photodamage [69]. From the perspective of the dynamic changes in gas exchange parameters, there are significant differences in the limiting factors of photosynthesis at different growth stages. At 30 days, both Gs and Ci decreased with the reduction in Mg concentration, indicating that photosynthetic limitation at this stage was primarily stomatal. In contrast, at later stages, although the Pn continued to decline, Ci increased, suggesting that photosynthesis became progressively constrained by non-stomatal factors. This pattern is consistent with the findings of Lin-Tong Yang et al., who reported similar responses in *Citrus sinensis* grown under Mg-deficient conditions [70]. The lower Chla/b ratio during 60–90 days (particularly in LM) likely reflects enhanced Chlb synthesis to develop a more efficient light-harvesting antenna for maximal light capture and rapid growth. Its subsequent increase at 120 days may represent an adaptive response to excess light, preventing damage to the photosynthetic apparatus. Elevated Car levels may correspond to stage-specific roles-early photoprotection and preferential chlorophyll degradation at later stages. At the same time, the antioxidant system is activated under low magnesium treatment [71]: MDA levels spike [72], and POD activity declines [73], further indicating oxidative stress. Enzymes essential to photophosphorylation, including Rubisco and Ca^2+^/Mg^2+^-ATPase, are also impaired [74]. Rubisco, a Mg-dependent enzyme [75], shows reduced activity under LM, though a temporary rise at day 60 may reflect compensatory accumulation. In addition, Mg deficiency affects SLA, LDMC, and pigment synthesis, shaped by both synergistic and antagonistic nutrient interactions [18,76]. Notably, elevated aluminum, a known phytotoxin, worsens Mg-related stress [77]. These combined perturbations reduce chlorophyll content, impair photosystem performance, and decrease carbon assimilation. Mg deficiency destabilizes ribosomes, impedes protein synthesis, and directly suppresses cell division, limiting the increase in leaf cell number [18]. Collectively, these findings suggest that magnesium deficiency disrupts key physiological processes in plants.

High magnesium concentration treatments (HM), on the other hand, had mixed effects. It enhanced Ci, Tr, and the activity of Rubisco and ATPase, delayed MDA accumulation [78], and extended leaf lifespan. However, as the most stress-sensitive component of the photosynthetic apparatus, PSII in the HM treatment likely exhibited an early detectable decline in efficiency at 60 days, as evidenced by the decrease in Fv/Fm [79]. The stress intensity at this stage remained relatively low, inducing only a slight numerical increase in POD activity without causing significant lipid peroxidation (no significant change in MDA), the photosynthetic system may already have experienced ionic imbalance or impaired photoprotective regulation under high Mg concentration. This could explain the seemingly contradictory phenomenon in HM, where certain photosynthetic parameters were enhanced while fluorescence parameters were inhibited. Such inconsistency may arise from imbalanced energy distribution mechanisms or dynamic ROS accumulation, leading to mismatched electron transport and photochemical quenching capacity. Consequently, accelerated electron flow without adequate energy dissipation renders PSII more susceptible to photooxidative damage. Overall, while moderate Mg levels promote photosynthesis and growth [80,81,82], excess input may destabilize the photosynthetic system. These findings highlight the necessity of precise regulation of Mg nutrition, balancing between deficiency and oversupply to sustain photosynthetic efficiency and ensure the healthy growth of C. lanceolata.

### 4.3. Growth Characteristics and Regulatory Factors of Mature Leaves in Cunninghamia lanceolata Seedlings Exposed to Magnesium Stress

The structural disintegration of cells serves as a critical indicator of the physiological status of mature leaves in *Cunninghamia lanceolata* seedlings subjected to prolonged magnesium stress. This phenomenon is observed under both magnesium deficiency and excess conditions [40,83,84]. In the present investigation, MDA content occupies a central role within both structural equation models, where it exhibits consistent directional effects across all treatment conditions (Figure 6a, Figure 8b,c). This parameter integrates key characteristics of mature leaf growth, reflecting two fundamental processes: energy conversion and material accumulation. From the perspective of photosynthetic physiology, MDA content demonstrates a strong correlation with the activities of Ca^2+^/Mg^2+^-ATPase and Rubisco, two enzymes pivotal in light energy conversion and carbon assimilation. These enzymes show a pronounced inverse relationship with dry matter accumulation in mature leaves (Figure 8b). The negative impact of Ca^2+^/Mg^2+^-ATPase activity may be attributed to the energy consumption associated with stress-induced physiological responses [85], whereas the positive influence of Rubisco highlights the critical contribution of carbon fixation to dry matter accumulation. Abnormal expression of Rubisco-related genes and a decline in carboxylation rate under magnesium stress have been documented in various species, including banana [86], tomato [36], and tea [85]. Nevertheless, our analysis revealed no significant correlation between leaf carbon content and dry matter accumulation. In contrast, Papadakis, I.E. et al. reported the accumulation of non-structural carbohydrates in the leaves of *Citrus sinensis* cv. Newhall under magnesium deficiency [87], suggesting that labile carbon components might mitigate the adverse effects of impaired carbon fixation [88]. Furthermore, MDA content serves primarily as a marker of aluminum ion fluctuations within the leaf dry matter system, which are influenced by nutrient element accumulation under stress conditions. This alteration may be driven by the cell’s tendency to sequester toxic elements such as aluminum in aging or damaged vacuoles [89], which subsequently affects calcium accumulation in leaves through competitive binding at available sites [90]. In addition, the synergistic relationship between magnesium and potassium has been well-established in the literature. Excess magnesium increases potassium concentration in buds, while a negative correlation between these two elements is observed in leaves and roots [91]. This finding corroborates our results (Figure 8c) and underscores the general principle that mobile elements are preferentially allocated to growth points [92]. Papadakis, I.E. et al. proposed that phosphorus absorption is not influenced by magnesium efficiency [87], and our structural equation modeling further supports this by indicating that the relationship between magnesium treatments and phosphorus content is not statistically significant (Figure 8c). Nonetheless, phosphorus, like nitrogen and potassium, remains a crucial regulator of dry weight in mature leaves. In conclusion, our findings suggest that photosynthetic enzyme activities, the availability of mobile nutrients, and calcium content are pivotal determinants influencing the growth of mature leaves under magnesium stress in seedlings.

### 4.4. Practical Implications for C. lanceolata Plantation Management

*C. lanceolata* is one of China’s most important timber species. However, repeated monoculture has degraded soil fertility and reduced site productivity [93]. To reverse this trend, better nutrient cycling is needed, including practices like retaining harvest residues and maintaining understory vegetation to improve organic matter inputs [94].

Magnesium should be prioritized alongside nitrogen, phosphorus, and potassium in fertilization strategies [95,96]. Our findings show that low magnesium concentration impairs leaf development, photosynthetic efficiency, and chloroplast stability, underlining the need for supplementation. Regular soil testing and targeted Mg application should be incorporated into plantation management. Both foliar spraying and root-zone application are effective, and timing fertilization to match developmental stages can boost nutrient uptake [18,97]. Implementing these practices can improve photosynthesis, increase biomass yield, and support long-term soil fertility-ensuring more sustainable *C. lanceolata* plantation management.

### 4.5. Limitations

This study offers preliminary insights into the physiological responses and structural adaptations of photosynthetic tissues in mature *C. lanceolata* leaves under varying magnesium (Mg) supply conditions. However, several limitations warrant attention to improve future research in this area. A primary constraint was the limited sample size [98], which restricted the analytical depth and reduced the robustness of our conclusions. Given Mg’s essential role in photosynthesis, energy metabolism, and nutrient cycling, disruptions in its availability trigger complex physiological adjustments. The small dataset necessitated the use of a least squares model, which may have compromised the accuracy and generalizability of our results. Another notable limitation lies in the modeling framework, which did not adequately incorporate latent variables. This shortfall likely stems from both the sample limitations and the intrinsic characteristics of the measured parameters [98,99]. Mg deficiency influences multiple physiological pathways, with responses shaped by both immediate environmental fluctuations and cumulative stress effects. In our approach, we constructed separate models for photosynthetic and nutrient-related traits. Photosynthetic responses tend to vary with short-term environmental conditions (e.g., light intensity, air movement), whereas nutrient dynamics often reflect longer-term processes. While this distinction offered some analytical clarity, it may have overlooked important cross-system interactions and feedbacks. Future work should aim to refine modeling strategies by incorporating integrated variables that reflect both immediate and cumulative physiological responses. Increasing the sample size will also be critical to enhance statistical power and model reliability. Additionally, because Mg is a highly mobile nutrient, its redistribution and internal reutilization under stress conditions should be explored more thoroughly. Finally, this study focused exclusively on mature leaves, neglecting younger leaves and meristematic tissues, which are often more responsive to nutrient stress. Addressing this gap will be essential for developing a more comprehensive understanding of Mg dynamics in plant development.

## 5. Conclusions

This study demonstrates that magnesium (Mg) supply levels regulate the photosynthetic performance and leaf functional traits of *C. lanceolata* seedlings. LM causes magnesium stress, disrupts chloroplast ultrastructure, impairs photochemical processes, and significantly reduces the net photosynthetic rate (Pn), accompanied by a decline in Fv/Fm below 0.8 and inhibition of Rubisco and Ca^2+^/Mg^2+^-ATPase activities, ultimately altering key growth traits and lowering leaf dry matter content (LDMC). In contrast, moderate Mg supply enhances intercellular CO_2_ concentration (Ci), transpiration rate (Tr), and the activities of Rubisco and ATPase while delaying MDA accumulation. However, it slightly decreases Fv/Fm. Overall, both high and low magnesium concentration treatments negatively influence leaf photosynthesis, with LM exerting a more pronounced inhibitory effect. Magnesium plays a complex regulatory role in plant physiology. However, the underlying signaling pathways and the roles of magnesium transporters remain largely unknown. Future studies using multi-omics approaches—such as transcriptomics and proteomics—are needed to identify the key genes and regulatory networks involved in magnesium stress responses.

## Figures and Tables

**Figure 1 plants-14-03542-f001:**
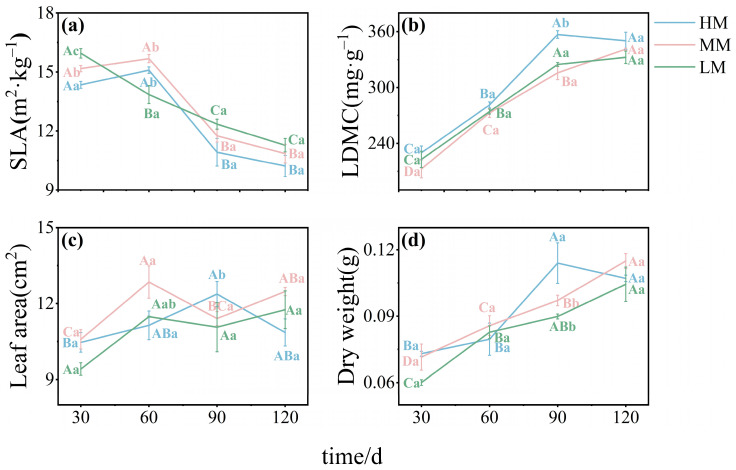
**Leaf morphological traits of *C. lanceolata* under different magnesium treatments.** Panels (**a**–**d**) show specific leaf area (SLA), leaf dry matter content (LDMC), leaf area, and leaf dry weight, respectively. SLA was calculated as leaf area divided by dry weight. The *x*-axis shows sampling time (30, 60, 90, and 120 days). Values are means ± SE (n = 20). Uppercase letters indicate significant differences across time points within a treatment, and lowercase letters indicate differences among treatments at the same time (*p* < 0.05).

**Figure 2 plants-14-03542-f002:**
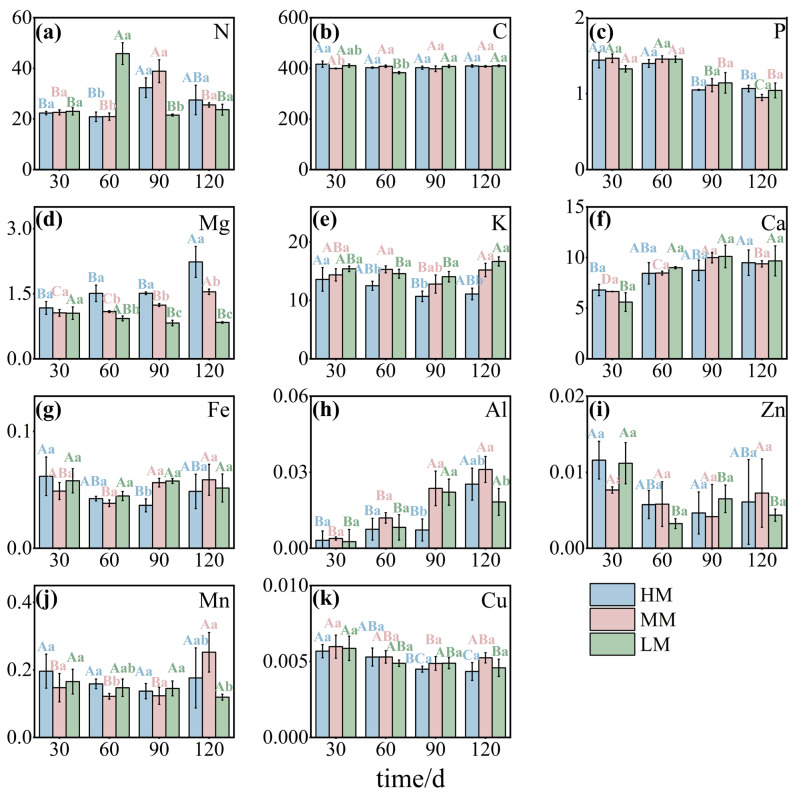
**Concentrations of mineral elements in *C. lanceolata* leaves under different magnesium treatments.** Panels (**a**–**k**) represent nitrogen (N), carbon (C), phosphorus (P), magnesium (Mg), potassium (K), calcium (Ca), iron (Fe), aluminum (Al), zinc (Zn), manganese (Mn), and copper (Cu), expressed as mg·g^−1^. Values are means ± SE (n = 20). Uppercase letters indicate significant differences across time points within a treatment, and lowercase letters indicate differences among treatments at the same time (*p* < 0.05).

**Figure 3 plants-14-03542-f003:**
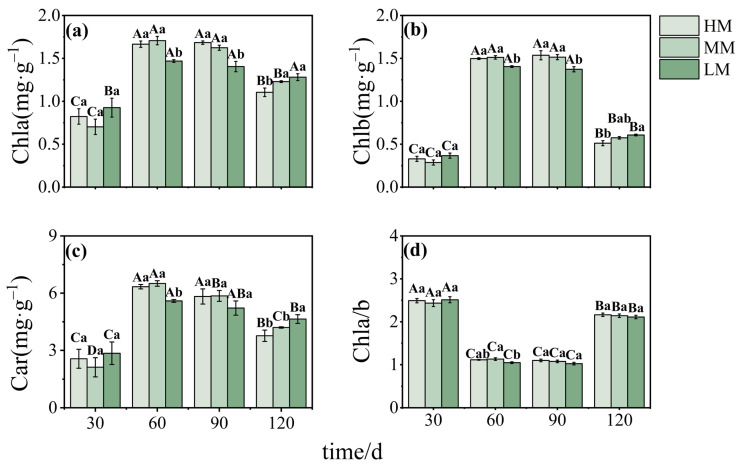
**Photosynthetic pigments in *C. lanceolata* leaves under different magnesium treatments.** Panels (**a**–**d**) show Chl a, Chl b, carotenoids, and the Chl a/b ratio. Values are means ± SE (n = 20). Uppercase letters indicate significant differences across time points within a treatment, and lowercase letters indicate differences among treatments at the same time (*p* < 0.05).

**Figure 4 plants-14-03542-f004:**
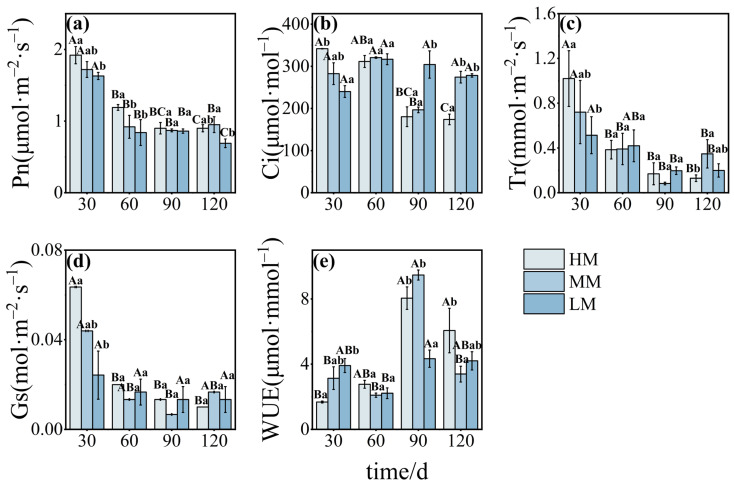
**Photosynthetic parameters in *C. lanceolata* under magnesium treatments.** Panels (**a**–**e**) show Pn, Ci, Tr, Gs and WUE. Values are means ± SE (n = 20). Uppercase letters indicate significant differences across time points within a treatment, and lowercase letters indicate differences among treatments at the same time (*p* < 0.05).

**Figure 5 plants-14-03542-f005:**
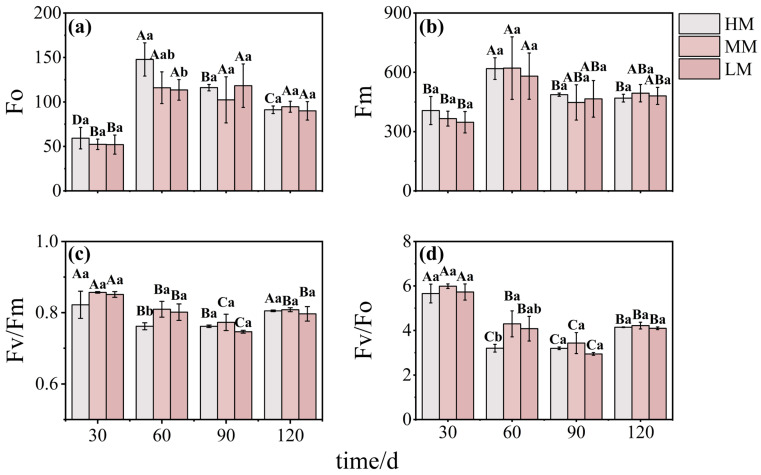
**Chlorophyll fluorescence traits in *C. lanceolata* under magnesium treatments.** Panels (**a**–**d**) show Fo, Fm, Fv/Fm, and Fv/Fo. Values are means ± SE (n = 20). Uppercase letters indicate significant differences across time points within a treatment, and lowercase letters indicate differences among treatments at the same time (*p* < 0.05).

**Figure 6 plants-14-03542-f006:**
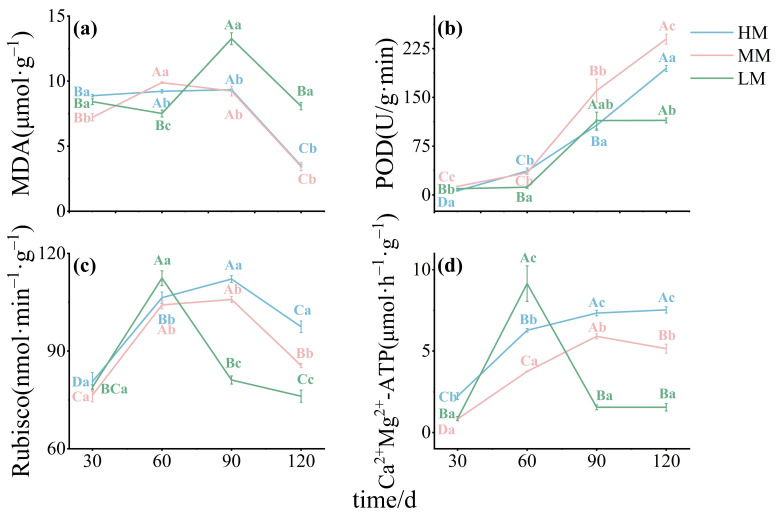
**Antioxidant and photosynthetic enzyme activities in *C. lanceolata* leaves under magnesium treatments.** Panels (**a**–**d**) show MDA, POD, Rubisco, and Ca^2+^Mg^2+^-ATPase. Values are means ± SE (n = 20). Uppercase letters indicate significant differences across time points within a treatment, and lowercase letters indicate differences among treatments at the same time (*p* < 0.05).

**Figure 7 plants-14-03542-f007:**
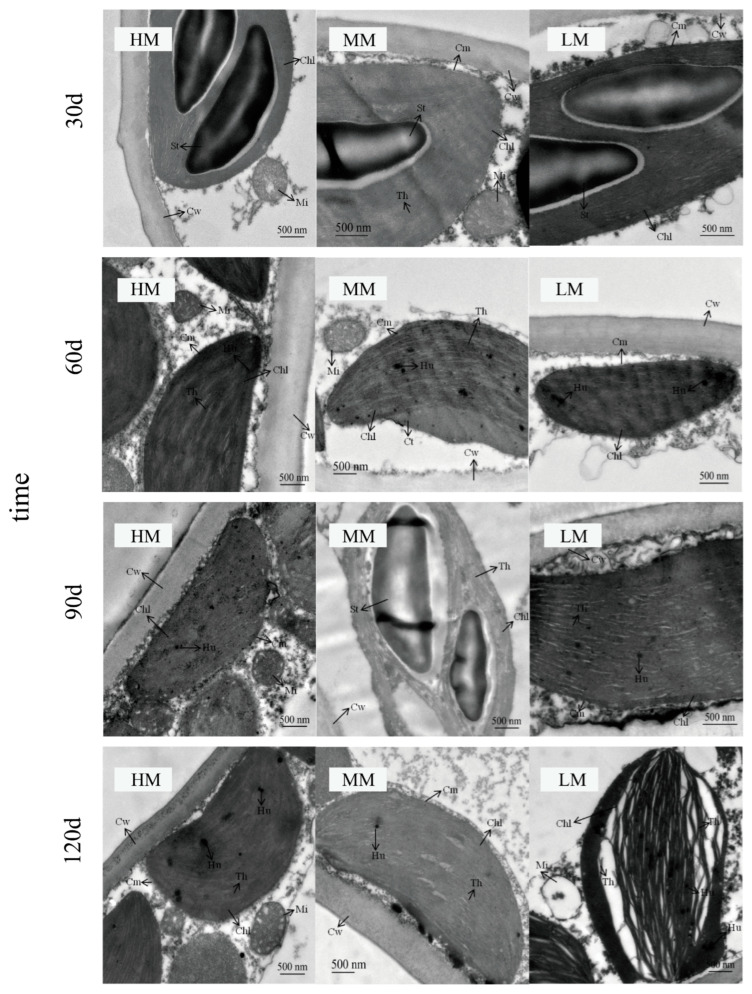
**Chloroplast ultrastructure in newly developed leaves of *C. lanceolata* under magnesium treatments.** St = starch grain, Hu = osmiophilic granule, Mi = mitochondrion, Chl = chloroplast, Cw = cell wall, Cm = chloroplast membrane, Th = thylakoid, Nu = nucleus.

**Figure 8 plants-14-03542-f008:**
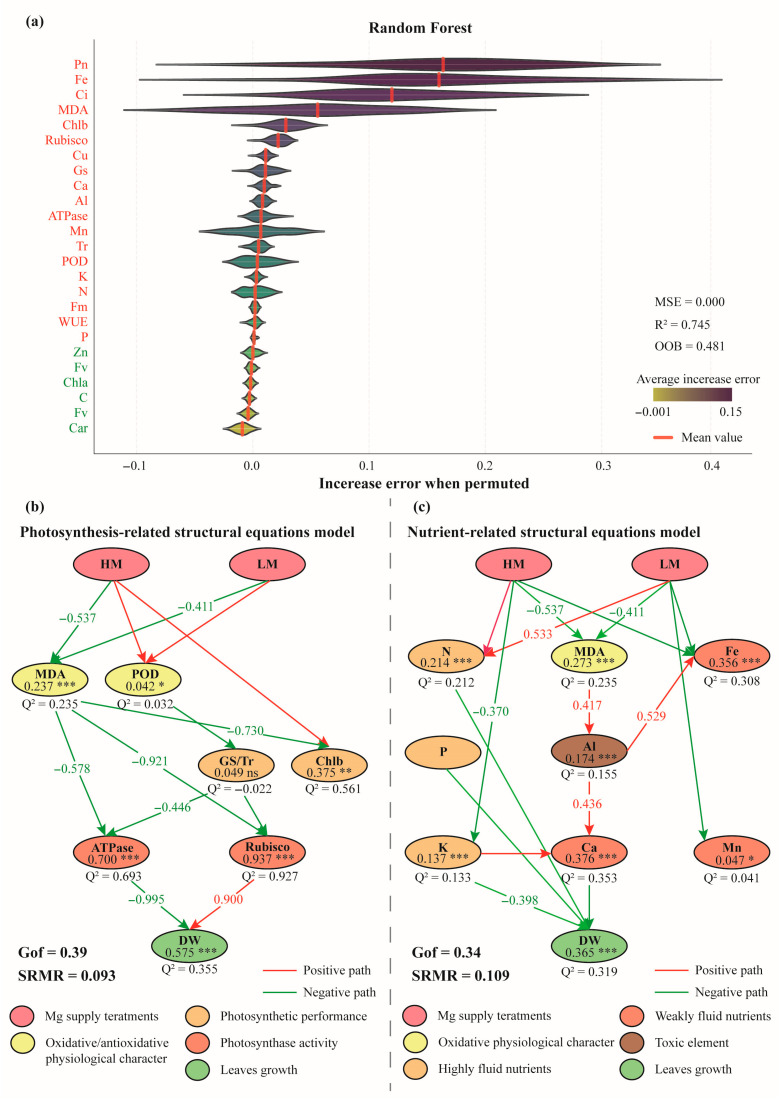
**Structural equation model depicting the multi-indicator importance assessment and the influence of divergent magnesium treatments on mature-leaf growth.** (**a**) The model error generated after each feature is randomly permutation 30 times under the random forest model trained with out-of-bag (OOB) data. Colors of the features listed left of the panel represent whether the increase error when permuted > 0.0 (red) or <0.0 (green). (**b**) Photosynthesis-related structural equations model. (**c**) Nutrient-related structural equations model. Arrows indicate potential pathways; path coefficients greater than 0.400 are displayed along the corresponding paths to denote relative influence strength. The R^2^ and significance level (ns indicates *p* > 0.05; * indicates *p* < 0.05; ** indicates *p* < 0.01; *** indicates *p* < 0.001) of each variable are presented beneath the variable name, whereas the blindfolding Q^2^ value is shown below the variable node. Colors of nodes and paths are used to differentiate variable and pathway categories.

## Data Availability

The original contributions presented in this study are included in the article. Further inquiries can be directed to the corresponding author.

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
