# Peer review of "Responses to Different Magnesium Supply Treatments in the Mature Leaves of *Cunninghamia lanceolata* Seedlings: Morphological, Physiological, and Structural Perspectives"

_plants, 2025, doi:10.3390/plants14223542_

Round 1

Reviewer 1 Report

Comments and Suggestions for Authors

The manuscript systematically investigates the effects of different Mg levels on leaf morphology, physiology, and ultrastructure in fir seedlings. The data dimensions are rich, covering growth, photosynthetic physiology, enzyme activity, and cellular ultrastructure, and it details the adverse effects of both Mg deficiency and excess on fir. However, the manuscript has significant deficiencies in experimental design, method description, result interpretation, data processing, and discussion, which obscure the clarity and impact of its scientific contribution.

Specific Major Concerns:

1. Title, Abstract, and Keywords
1)    The manuscript type (e.g., Article, Review) must be correctly selected, not copied directly from the template placeholder ("Type of the Paper...").
2)    No keywords are provided. Please add.
3)    The experimental material used is one-year-old seedlings, which differ significantly from mature trees. It is recommended to define and specify this scope in both the Title and Abstract.

2. Introduction
1)    The effects of Mg deficiency and excess on plants are already well-established. The manuscript's innovation needs to be more clearly refined.
2)    Some content in the Introduction is either general knowledge or has low relevance to the manuscript's research focus. It is recommended to thoroughly reorganize the literature review, focusing sharply on the manuscript's specific research content, measured parameters, and main conclusions.

3. Methods
1)    As the experiment was conducted in an artificial climate chamber with fully controlled environmental conditions, the description of the local natural climate is unnecessary and should be removed.
2)    How were the three Mg treatments implemented? Relevant details must be explicitly stated in the Methods section; providing only concentration levels is insufficient.
3)    The nutrient solution described contains Mg. Is this concentration sufficient to meet the seedlings' Mg needs? The experimental design has a clear flaw: either a true Mg-free control should be included, or the background Mg requirement for normal seedling growth must be explicitly stated, upon which the treatment concentration gradients are based.
4)    Could daily irrigation with sodium ions cause salt stress? This potential confounding factor needs addressing.
5)    The replication setup for the three treatments and the specific experimental design used are not described. Please provide this information.
6)    The method describes pooling "12 leaves... into three biological replicates," meaning each replicate is a composite sample of four leaves. This weakens the statistical meaning of biological replication, as it does not reflect inter-individual variation. True biological replicates should come from different, independent individual plants.
7)    The experiment lasted only 120 days, providing limited support for the issues raised in the Abstract regarding "repeated planting cycles and monoculture." It is recommended to refine the stated significance of the research to better match the experimental scale.
8)    The analysis of "Mineral nutrients" should not be categorized under "Photosynthetic physiology detection." Furthermore, the title "Key metabolites and antioxidant activity analysis" inaccurately represents the included indicators (MDA, POD). Please correct the categorization and titles.
9)    The Transmission Electron Microscopy analysis requires more detailed parameters (e.g., specific fixation times, resin embedding details, staining conditions) to allow for reproducibility.

4. Results
1)    The conclusion that both deficiency and excess have adverse effects seems "obvious." To make the research description more objective, consider renaming the treatments "Low Mg concentration," "Medium Mg concentration," and "High Mg concentration," rather than subjectively labeling them as deficient and excess. The current presentation gives the impression that the results were assumed before the experiment began.
2)    The complex dynamics observed in the SM treatment (e.g., SLA initially higher then lower, LDMC increase) likely represent physiological disruption under stress rather than an adaptive strategic shift. The authors need to re-examine and precisely their conclusions accordingly.
3)    Figures 1 and 6 lack error bars. Please add them and specify in the figure legends what they represent (e.g., SD, SE).
4)    There are logical contradictions in interpreting the path coefficients from the Structural Equation Modeling (SEM). For instance, the authors state "Mg treatment had negative effects on nutrient elements," but Mg treatment level is the independent variable, and the "nutrient elements" latent variable includes Mg content. A reasonable model should show a positive path coefficient here (higher Mg treatment -> higher Mg content). This apparent contradiction must be clarified; it might stem from incorrect latent variable construction or model specification.
5)    A table is needed detailing which specific measured indicators constitute each latent variable (e.g., "Nutrient elements," "Enzyme activity") and their respective loadings, to justify the latent variable construction.
6)    Reporting only GoF=0.53 is insufficient. Provide additional model fit indices (e.g., SRMR, NFI) to demonstrate model validity.
7)    The changes in element concentrations are complex. The text should highlight the most critical changes rather than describing all elements with equal emphasis.

5. Discussion and Conclusions
1)    The Discussion cites literature extensively, which helps connect the results to existing knowledge. However, it lacks in-depth mechanistic speculation for the observed phenomena (e.g., initial SLA increase under SM; decrease in Fv/Fo under EM). For example: Is the initial SLA increase under SM due to greater inhibition of cell division compared to cell expansion, or passive thinning due to water stress? What causes the contradictory phenomenon in EM where some photosynthetic parameters are "promoted" while fluorescence parameters are "inhibited"? Is it related to photoinhibition, ROS accumulation, or disrupted ion balance? Incorporate more plausible speculative explanations based on physiological and biochemical knowledge.
2)    The Discussion completely fails to utilize the SEM results to support its arguments. The SEM, intended to represent the core findings synthesized from the data, should be a central focus in the Discussion. For example, explain how Mg supply ultimately affects growth through which specific pathways identified by the model.
3)    The Conclusions should be consolidated into a single paragraph and include quantitative descriptions of key results. The Perspectives section should avoid straying too far from the study's direct findings.

Comments on the Quality of English Language

The language is generally readable but contains minor grammatical inaccuracies and repetitive phrasing. Professional language editing is recommended. For instance, the frequent use of "show" ("showed," "shown") could be replaced with synonyms like "exhibit," "display," or "indicate" to enhance variety.

Author Response

Thank you very much for your valuable comments on our manuscript. We have revised the manuscript in accordance with your suggestions. Please refer to the attached point-by-point responses.

Reviewer 2 Report

Comments and Suggestions for Authors

The manuscript deals with a relevant subject to plants. The ms is very interesting, with an interesting set of data.However, there are areas in the ms which require improvement and clarification, as presented below. Thus, I recommend that the manuscript should be accepted after major revision.

Specific points:

  1. Line 98: photosynthetic photon flux density instead of light intensity.
  2. Title 2.6. Please, change the word detection (is not the most appropriate term).
  3. Section 2.6 - Reference for photosynthetic pigments calculation should be provided.
  4. Information about the time of the day of leaf gas exchange measurements should be included.
  5. References relatively to 2.7 analysis, when applied, should be provided.
  6. Figure 1 (and in other situations): Units inside parenthesis.
  7. How authors explain such high levels of K, Ca, Al and Mn? Are there no mistakes?
  8. Legend Figure 2 (and in other situations): Standard errors and statistical analysis should be included in legend.
  9. Figure 3: Units by fresh or dry weight? Information should be provided.
  10. How authors explain such low Cha/Clb values at 60 and 90 days and thereafter the significant increase at 120 days? Readers want know in the discussion section.
  11. How authors explain higher values of carotenoids than chloropylls? Readers want know in the discussion section.
  12. Authors should provide results of stomatal conductance.
  13. How authors explain, at 60 days, high Pn in EM treatment notwithstanding the low Fv/Fm ratio? Readers want know in the discussion section.
  14. Line 302: main photosynthetic organ instead of main sites of photosynthesis.
  15. When are detected lower values of Pn among treatments or decreases of Pn values among dates, will have been verified stomatal or non-stomatal limitations to photosynthesis? Readers want know in the discussion section.

Author Response

(The authors gave the same response as above.)

Round 2

Reviewer 1 Report

Comments and Suggestions for Authors

This revised manuscript has addressed all my comments comprehensively and satisfactorily. The authors have not only meticulously revised textual expressions, figures, and descriptions of experimental methods but have also provided convincing explanations and substantive improvements regarding key scientific issues (such as the rationality of the experimental design, control of potential confounding factors, and reconstruction of the structural equation model). The scientific rigor and logical coherence of the paper have been significantly enhanced. It is recommended for acceptance.

Author Response

Thank you very much for your positive evaluation and constructive comments on our revised manuscript! We greatly appreciate your meticulous review and recognition of the improvements we made. Your affirmation of the enhanced scientific rigor and logical coherence of the paper has given us great encouragement. We will continue to uphold the rigorous attitude of scientific research in subsequent work. Once again, we would like to express our sincere gratitude to you for your valuable time and professional guidance, which have played a crucial role in improving the quality of this research.

Reviewer 2 Report

Comments and Suggestions for Authors

The manuscript has been significantly improved. However, the units for stomatal conductance (figure included in this version of the manuscript) are incorrect. Therefore, the manuscript should be accepted after minor revision.

Author Response

Comments 1:The units for stomatal conductance (figure included in this version of the manuscript) are incorrect.

Response 1:We appreciate you pointing out the error in the units for stomatal conductance. After carefully reviewing our data files and cross-referencing with standard conventions in published literature, we confirm that the units were incorrectly reported in the previous version. We have now corrected this in the revised manuscript, and the stomatal conductance values are presented with the standard units (mol·m⁻²·s⁻¹). Please see the updated figure (Figure 4) and corresponding figure legend for details. We apologize for any confusion this may have caused and thank you again for your meticulous review, which has helped improve the accuracy of our work.